# Impact of Whole Genome Doubling on Detection of Circulating Tumor DNA in Colorectal Cancer

**DOI:** 10.3390/cancers15041136

**Published:** 2023-02-10

**Authors:** Jonas Kabel, Tenna Vesterman Henriksen, Christina Demuth, Amanda Frydendahl, Mads Heilskov Rasmussen, Jesper Nors, Nicolai J. Birkbak, Anders Husted Madsen, Uffe S. Løve, Per Vadgaard Andersen, Thomas Kolbro, Alessio Monti, Ole Thorlacius-Ussing, Mikail Gögenur, Jeppe Kildsig, Nis Hallundbæk Schlesinger, Peter Bondeven, Lene Hjerrild Iversen, Kåre Andersson Gotschalck, Claus Lindbjerg Andersen

**Affiliations:** 1Department of Molecular Medicine, Aarhus University Hospital, 8200 Aarhus, Denmark; 2Department of Clinical Medicine, Aarhus University, 8000 Aarhus, Denmark; 3Department of Surgery, Regional Hospital Herning, 7400 Herning, Denmark; 4Department of Surgery, Regional Hospital Viborg, 8800 Viborg, Denmark; 5Department of Surgery, Odense University Hospital, 5000 Odense, Denmark; 6Department of Surgery, Odense University Hospital, 5700 Svendborg, Denmark; 7Department of Surgery, Hjørring Hospital, 9800 Hjørring, Denmark; 8Clinical Cancer Research Center, Aalborg University, 9000 Aalborg, Denmark; 9Center for Surgical Sciences, Zealand University Hospital, 4600 Køge, Denmark; 10Department of Surgery, Copenhagen University Hospital, 2730 Herlev, Denmark; 11Department of Surgery, University Hospital Bispebjerg, 2400 Copenhagen, Denmark; 12Department of Surgery, Regional Hospital Randers, 8900 Randers, Denmark; 13Department of Surgery, Aarhus University Hospital, 8200 Aarhus, Denmark; 14Department of Surgery, Regional Hospital Horsens, 8700 Horsens, Denmark

**Keywords:** whole genome doubling, circulating tumor DNA, colorectal cancer, cancer diagnostics

## Abstract

**Simple Summary:**

Measurements of circulating tumor DNA is a promising new tool in early detection of cancer and the detection of residual cancer disease. To ensure a high test sensitivity, detailed knowledge about factors that may influence circulating tumor DNA levels is needed. To this end, we investigate a cancer phenomenon in which tumor cells undergo a doubling of the entire genome, thus containing double the normal amount of DNA. We test the hypothesis that DNA from these genome-doubled tumors is more easily detected in blood plasma samples compared to non-genome-doubled tumors. We found that the probability of detecting circulating tumor DNA was higher among genome-doubled tumors than non-genome-doubled tumors. The results thus indicate that increased amounts of tumor-cell DNA lead to increased amounts of tumor DNA in the circulation.

**Abstract:**

Objective: Circulating tumor DNA (ctDNA) is a candidate biomarker of cancer with practice-changing potential in the detection of both early and residual disease. Disease stage and tumor size affect the probability of ctDNA detection, whereas little is known about the influence of other tumor characteristics on ctDNA detection. This study investigates the impact of tumor cell whole-genome doubling (WGD) on the detection of ctDNA in plasma collected preoperatively from newly diagnosed colorectal cancer (CRC) patients. Methods: WGD was estimated from copy numbers derived from whole-exome sequencing (WES) data of matched tumor and normal DNA from 833 Danish CRC patients. To explore if tumor WGD status impacts ctDNA detection, we applied tumor-informed ctDNA analysis to preoperative plasma samples from all patients. Results: Patients with WGD+ tumors had 53% increased odds of being ctDNA positive (OR = 1.53, 95%CI: 1.12–2.09). After stratification for UICC stage, the association persisted for Stage I (OR = 2.44, 95%CI: 1.22–5.03) and Stage II (OR = 1.76, 95%CI: 1.11–2.81) but not for Stage III (OR = 0.83, 95%CI: 0.44–1.53) patients. Conclusion: The presence of WGD significantly increases the probability of detecting ctDNA, particularly for early-stage disease. In patients with more advanced disease, the benefit of WGD on ctDNA detection is less pronounced, consistent with increased DNA shedding from these tumors, making ctDNA detection less dependent on the amount of ctDNA released per tumor cell.

## 1. Introduction

Measurements of circulating tumor DNA (ctDNA) are currently being investigated as a tool in early detection and postoperative monitoring of cancer [1,2,3,4,5,6,7,8,9]. High test sensitivity is required for successful integration into clinical practice, and to this end, knowledge about tumor characteristics affecting ctDNA detection is beneficial. Several factors have been reported to affect tumor shedding and, thus, ctDNA levels. Among these are the Union of International Cancer Control (UICC) stage, tumor size, tumor cell turnover, and tumor mutational burden [10,11,12,13,14,15,16]. In addition to tumor shedding, we hypothesize that the number of DNA copies per tumor cell is also likely to influence the probability of ctDNA detection. The more DNA copies released per tumor cell, the higher the level of ctDNA in the blood and thus increased likelihood of detection. Large-scale amplification events, such as a mitotic cytokinesis defect leading to whole-genome doubling (WGD), will increase the number of DNA copies per tumor cell [17,18,19]. WGD frequently occurs in human cancers [19,20], fuels tumor development [20], and its presence is a predictor of poor prognosis in colorectal cancer (CRC) patients [17].

In this study, we aim to explore the impact of tumor WGD on the probability of detecting ctDNA in CRC patients. Furthermore, we investigate whether this association is affected by the UICC stage and tumor histopathology.

## 2. Methods and Materials

### 2.1. Patient Selection

The study cohort consisted of Danish UICC stage I-III CRC patients undergoing curative-intent surgery between 2014 and 2021 at one of the following 12 Danish hospitals: Aarhus University Hospital, Odense University Hospital, Aalborg University Hospital, Regional Hospital Herning, Regional Hospital Viborg, Regional Hospital Randers, Regional Hospital Horsens, Bispebjerg Hospital, Herlev Hospital, Svendborg Hospital, Hjørring Hospital, Køge Hospital. The inclusion criteria were (1) UICC stage I-III disease, (2) ctDNA measured in plasma preoperatively, and (3) whole exome sequencing of the primary tumor and matched normal DNA. The study has been approved by the Committees on Biomedical Research in the Central Region of Denmark (1-10-72-3-18 and 1-10-72-223-14). All patients provided written informed consent, and the study was performed in accordance with the declaration of Helsinki.

### 2.2. Sample Collection

Tumor tissue samples were collected from the resected primary tumor (fresh frozen (FF) or formalin-fixed and paraffin-embedded tissue (FFPE)). Blood samples were collected in K2-EDTA 10 mL tubes (Becton Dickinson, Franklin Lakes, NJ, USA) 0–28 days (median 6 days) before surgery. Plasma was isolated from whole blood within 2 h by double centrifugation at 3000× *g* (each 10 min). Peripheral blood mononuclear cells (PBMCs) were collected after the first centrifugation. Blood-derived samples were stored at −80 °C until DNA extraction.

### 2.3. DNA Extraction

Tumor DNA was extracted from FF tumor tissue samples using the Puregene DNA purification kit (Gentra Systems, Minneapolis, MN, USA) and from FFPE samples using the QiAamp DNA FFPE tissue kit (Qiagen, Hilden, Germany). PBMC DNA was extracted using the Qiasymphony DNA mini kit (Qiagen). Tumor and PBMC DNA were quantified by the Qubit™ dsDNA BR Assay Kit (Thermo Fisher, Waltham, MA, USA).

Cell-free DNA (cfDNA) was purified from 8 mL of plasma in a single replicate using the QIAamp Circulating Nucleic Acids kit (Qiagen) or the QIAsymphony DSP Circulating DNA Kit (Qiagen) on the QIAsymphony robot (Qiagen). The cfDNA was eluted in a 60 μL Suspension Buffer (Sigma-Aldritch, Saint-Louis, MO, USA). No extraction blanks were used. cfDNA was quantified by digital droplet PCR (ddPCR; Bio-Rad Laboratories, Hercules, CA, USA), with assays targeting regions on chr3 and chr7 with little copy-number variation in CRC, as described previously [21]. The cfDNA was frozen immediately and stored at −80 °C until use.

### 2.4. Whole Exome Sequencing

Sequencing libraries were constructed using the Twist Library Preparation kit after enzymatic fragmentation (TWIST Bioscience, San Francisco, CA, USA). Libraries were prepared with xGen UDI-UMI adapters (Integrated DNA Technologies (IDT), Inc., Coralville, IA, USA). Libraries with PBMC DNA, FF tissue DNA, and FFPE tissue DNA samples were prepared using 50 ng input DNA and either 10 min (PBMC and FF) or 6 min (FFPE) fragmentation. Library amplification was executed with 7 or 8 cycles of PCR. Exome hybridization capture was performed using the NGS Human Core Exome (TWIST Bioscience, ~33 Mb). Target-enriched libraries were sequenced on the NovaSeq platform with 2 × 150 bp paired-end sequencing to a targeted sequencing depth of 60× for PBMC DNA, 130× for FF tissue DNA, and 150× for FFPE tissue DNA.

Raw sequencing reads were converted into FastQ files using Illumina bcl2fastq. Sequencing adapters were removed bioinformatically by cutadapt (v3.0) [22], and trimmed reads were mapped to the human reference genome (hg38) using BWA-MEM (v0.7.17) [23]. Alignment was further processed as previously described [24]. Single-nucleotide polymorphism (SNP) concordance between tumor and matched PBMC DNA samples was examined to guard against sample swaps. This was carried out by the assessment of ~1200 exonic SNP sites with high inter-patient genotype diversity. For each patient, the genotype of all SNP sites from tumor and normal samples were compared to confirm intra-patient sample coherence. Somatic single nucleotide variants (SNVs) and small insertions/deletions were identified using GATK Mutect2 [25]. Variant clonality was assessed using Bubbletree [26] and pureCN [27,28]. Both tools use the WES data to estimate the purity of the analyzed tumor biopsy, the tumor ploidy, and the variant allele-specific copy number. From these numbers, the variant allele frequency (VAF) is transformed into the cancer cell fraction (CCF), i.e., the fraction of tumor cells carrying a particular variant. Mutations with a CCF lower than 0.9 were considered subclonal.

### 2.5. Detection of ctDNA

Two tumor-informed methods were used for ctDNA detection. Either ddPCR targeting a clonal mutation or deep targeted cfDNA sequencing. The latter targeted a fixed panel of 12 genes frequently mutated in CRC (Appendix A). Initially, the list of clonal mutations for each patient was compared to our in-house panel of >100 ddPCR mutation assays, as described previously [29]. The patient’s plasma sample was analyzed by ddPCR if an overlap was identified. If not, it was analyzed by deep targeted sequencing.

### 2.6. Deep Targeted cfDNA Sequencing

For the targeted cfDNA sequencing, libraries were prepared using KAPA Hyper (Roche, Basel, Switzerland) with xGen UDI-UMI adapters (Integrated DNA Technologies, Inc.). Libraries were enriched for regions of interest in two consecutive captures using a custom panel (Appendix A). This panel was designed based on data from The Cancer Genome Atlas Program (TCGA) [30]. The captured libraries were paired-end sequenced (2 × 150 bp) on the NovaSeq platform (Illumina, San Diego, CA, USA). All libraries were sequenced to saturation (median unique depth 15,716, Appendix A), meaning that the entire complexity of the captured library was sequenced.

Reads were mapped to hg38 using BWA mem [31], and subsequently, UMIs were grouped with the UMItools group [32] with the options ‘--method = directional’ and ‘--edit-distance-threshold = 1’. UMI consensus generation was performed with fgbio CallMolecularConsensusReads and CallMolecularConsensusReads commands using default settings [33] and requiring a minimum of three reads in a UMI family (‘--min-reads = 3’). fgbio ClipBam was run to avoid counting overlapping bases from the same read pairs twice. Forward and reverse strand read counts were generated using pysamstats [34], which were finally used for variant calling using the AND model of the Bayesian beta-binomial (bbb) function in the deepSNV R package [35] using a panel of normal cfDNA samples from 46 healthy individuals as background. A sample was called positive if at least one mutation identified in the primary biopsy had a posterior for the alternative model below 0.05.

### 2.7. ddPCR ctDNA Analysis

Our ddPCR approach, including assay design, cycling optimization, and error correction, has been extensively described elsewhere [24,29]. Here, we describe the relevant points in brief. All ddPCR assays consisted of a single primer set amplifying the target region, one probe reporting the mutation, and another probe reporting the corresponding wild-type sequence (primer and probe sequences and PCR cycling conditions in Appendix A). Assays were checked for linearity and sensitivity by a 4-point dilution series of tumor DNA in a fixed background of 10 ng wild-type DNA (mutant allele frequencies of 1%, 0.3%; 0.1%; 0.03%). An assay-specific noise profile was generated for every assay by applying it to 95 fragmented PBMC DNAs. These were PBMC DNAs from 19 healthy donors, and each was analyzed at five different concentrations to cover the variation in cfDNA concentration seen in cancer patients.

All ddPCR reactions were carried out in a 20 µL volume. The cfDNA from 8 mL plasma was split between six ddPCR reactions to avoid oversaturating the droplets. In addition to cfDNA, each ddPCR setup included reactions with a no-template control (water), a tumor DNA positive control, a PBMC DNA negative control from the patient’s own PBMCs (also checking for clonal hematopoiesis of indeterminate potential (CHIP)), and a PBMC DNA negative control from a healthy donor. Droplets were generated by the Automated Droplet Generator (Bio-Rad) and read on the QX200™ Droplet Reader (Bio-Rad).

The CASTLE algorithm [24] was used to compare the observed plasma signal to the assay-specific noise profile, thereby statistically determining the ctDNA status of each sample. Samples were called ctDNA positive if the CASTLE *p*-value was ≤ 0.01 for the H_0_: mutation concentration = 0.

### 2.8. WGD Estimation

WGD status for each tumor sample was inferred from copy number information obtained from applying the Allele-Specific Copy-number Analysis of Tumors (ASCAT) software to the tumor WES data, as described and validated previously [36,37]. Prior to WGD analysis, all ASCAT copy number profiles generated from the tumor WES data were subjected to manual QC. This was performed because ASCAT has been reported to produce unreliable outputs when the input data is noisy. ASCAT over- and underestimates copy numbers in samples with high tumor heterogeneity (high subclonality) and low tumor purity (high normal cell admixture), respectively [37]. Examples of tumors excluded due to poor QC are provided in Appendix A.

From the matched tumor and PBMC WES data, ASCAT extracted B-allele frequencies (BAFs) and logR ratios (LRRs) for approximately 30,000 Single Nucleotide Polymorphisms (SNPs). The ASCAT output was segmented allele-specific copy numbers for the entire tumor genome. To estimate whether a sample had undergone WGD, the fraction of the tumor genome that had a major allele copy number (MCN) of two or above was calculated. An MCN of two can only occur if a doubling of at least one allele in the given region has taken place. If a large fraction of the tumor genome has an MCN of two or more, this suggests a large genomic duplication event and is thus indicative of WGD. As previously described, samples in which more than 50% of the tumor genome had an MCN of two or more were classified as WGD+, while all other samples were classified as WGD− [20].

The reliability of the MCN WGD estimation method was assessed by using an orthogonal approach, where samples with a cancer cell ploidy larger than 2.5–1.25× FracLOH were defined as WGD [38]. The FracLOH is the fraction of the tumor genome showing loss-of-heterozygosity (LOH). The overall concordance between the MCN method and the LOH method was 92%, Cohen’s Kappa = 0.83 (Appendix A), confirming the robustness of MCN WGD estimation approaches. Due to the high concordance between the two methods, results are presented for the MCN method only.

### 2.9. Statistics

Univariable and multivariable log-binomial regression analyses were applied to assess the effect size and statistical significance of associations between tumor characteristics and ctDNA status. The multivariable logistic regression was calculated containing the variables “WGD”, “UICC Stage”, “Tumor Type”, “Venous Invasion”, “Tumor Size”, “Tumor Location”, and “MMR Status”. Associations were calculated as odds ratios and reported with a 95% confidence interval. Continuous outcomes were reported as median with range. Results were classified as statistically significant at *p* < 0.05. In our study, the ddPCR and deep targeted sequencing approaches for ctDNA detection had a nearly identical ctDNA-positive rate (Appendix A), and therefore we do not stratify the results by detection method. All calculations were performed in RStudio with R v. 4.1.2 [39].

## 3. Results

### 3.1. Study Population

In total, 833 patients were included in the study. After quality control of the WES data, 132 patients were excluded due to low signal-to-noise ratios caused by either poor data quality (*n* = 20), low tumor purity (*n* = 68), or high intra-tumor heterogeneity (*n* = 44) (Figure 1). The final study cohort consisted of 701 patients. The characteristics of the final study cohort were no different from the original full cohort (Table 1). All data on the final study cohort are available in Appendix A.

### 3.2. WGD and Tumor Size Correlate with ctDNA Detection in Stage-Stratified Analysis

To identify tumor characteristics associated with ctDNA positivity, we compared clinicopathological and WGD information for ctDNA-positive and -negative patients (Table 2). Tumor stage, size, venous invasion, left colon location, and WGD were all significantly associated with ctDNA detection. When stratifying for stage, tumor size was associated with ctDNA detection across all stages, with stage II and stage III patients having a notably higher mean tumor size than stage I patients. The WGD rate was significantly higher for ctDNA-positive than ctDNA-negative stage I and stage II patients, but not stage III patients. The WGD rate was highest for ctDNA-positive stage I patients. As the impact of WGD on ctDNA detection for stage II patients appeared to be intercalated between that of stage I and stage III patients, we sought to further subdivide stage II patients according to pathological tumor stage (pT stage). For stage II patients with pT3 but not pT4 tumors, we found a significantly higher frequency of WGD among ctDNA-positive patients (Appendix A).

### 3.3. WGD Increases ctDNA Detection in a Multivariable Model

To assess and quantify how the individual tumor characteristics impact the probability of ctDNA detection after mutual adjustment, we applied multivariable regression analysis (Table 3). WGD status was observed to be an independent predictor of ctDNA detection (OR = 1.74, 95%CI: (1.20–2.52), *p* = 0.004). Additionally, tumor size, tumor location, and stage III disease were also independent predictors, while no associations were found for stage II disease, tumor histology, venous invasion, or MMR status. Among the categorical variables, WGD, stage III disease and location in left colon or rectum were observed to have the highest odds ratios.

## 4. Discussion

In this study, we investigated the impact of tumor WGD on ctDNA detection in plasma samples from CRC patients. In accordance with the hypothesis that the number of DNA copies per tumor cell affects the probability of ctDNA detection, we observed higher odds of a patient testing ctDNA positive if the tumor was WGD+. The increased likelihood of detecting ctDNA in plasma from patients with WGD+ tumors most likely reflects the higher ctDNA levels achieved when a WGD+ tumor cell releases its DNA to the circulation relative to a WGD− tumor cell. The association between WGD and ctDNA detection persisted after adjusting for other tumor characteristics. Interestingly, while WGD status impacted ctDNA detection in stage I and stage II patients, there was no effect in stage III patients. It is well established that patients with late-stage disease often have more ctDNA in the circulation than patients with early-stage disease [11,40], indicating that their tumors generally shed more DNA, which in turn makes ctDNA detection easier. Many features associated with late-stage disease, such as lymphovascular invasion, increased proliferation, increased apoptosis, increased necrosis, depth of invasion (pT3-4), lymph node involvement (pN > 0), increased size, and increased vascular density are biophysically associated with an increased likelihood of tumor DNA shedding. Accordingly, it can be argued that for late-stage patients, the WGD status will have less impact on ctDNA detection because the shedding rate, even of diploid tumors, is sufficiently high for the ctDNA level to pass the limit of detection (Figure 2). By contrast, in patients with early-stage tumors, and hence a lower shedding rate, a WGD event may be what is needed to bring the plasma ctDNA concentration above the limit of detection. As emerging improvements in ctDNA detection technology continue to increase ctDNA detection sensitivities, the impact of factors like WGD on ctDNA detection is likely to become even more localized to cases of low shedding.

In stage II patients, the effect of a WGD event on ctDNA detection was intermediate to the effect in stage I and stage III patients. We speculated if this, in part, could be due to the heterogeneity of stage II. Stage II comprises both tumors confined to the intestinal wall (pT3) and tumors with growth through the intestinal wall and sometimes even into neighboring tissues or organs (pT4). For pT3 patients, the analysis showed significant enrichment of WGD among the ctDNA-positive relative to the ctDNA-negative patients (54% vs. 38%, *p* = 0.008). By contrast, the analysis showed no WGD enrichment for pT4 ctDNA-positive patients. Both the ctDNA positive and negative pT4 patients had a high WGD rate (54% vs. 60%, *p* > 0.9). While we note that the pT4 sample size (*n* = 23) was smaller than the pT3 sample size (*n* = 313), the observations nevertheless indicate that pT4 tumors shed more tumor DNA than pT3, thereby reducing the importance of WGD.

To our knowledge, this study is the first to investigate and demonstrate the impact of WGD on ctDNA detection. The reported findings imply that in tumor-informed ctDNA analyses, the mutational targets should optimally be present in more than one copy per cancer cell, particularly if the aim is to detect ctDNA in low tumor burden settings, such as early cancer detection and minimal residual disease detection.

The study has several limitations. Despite being based on 701 patients, the main limitation was the modest number of patients remaining in the subgroups after stratification. The minor limitations include ctDNA detection. Since all plasma samples analyzed for this study were collected before tumor resection, ctDNA should, in theory, be present in all samples. Despite this, ctDNA was only detected in 63% of samples. Likely, this was because the ctDNA level in the patient was below the limit of detection for our ctDNA detection approaches. Low ctDNA levels can be due to certain tumor characteristics reducing the shedding capability of the tumor. Across all UICC stages, patients with detectable ctDNA were observed to have a significantly larger median tumor size compared to ctDNA-negative patients. Small tumor size thus likely accounts for some of the negative results, as supported by other studies [10,12,13,14,41]. Furthermore, both our ctDNA detection methods were targeted and tumor-informed. Hence, another factor possibly influencing ctDNA detection is target selection. Both methods are theoretically vulnerable to inadvertently targeting subclonal mutations. To mitigate this source of error, we performed an extensive and thorough clonality assessment prior to selecting targets. Another potential limitation is the lack of a best practice for WGD estimation. To overcome this, we used an objective WES-based approach, which previously has been verified [38], and furthermore, we confirmed our WGD calls using an independent approach. A final limitation was the use of “tumor’s largest diameter” as a proxy of size. The diameter is an imprecise measure of tumor mass. However, it was used because it was the only tumor size metric standardly measured on all patients. Despite the limitations, we believe that our cohort size and rigorous data analysis approach have yielded robust results, which are broadly applicable to the ctDNA research field.

## 5. Conclusions

In this study, we investigated the impact of tumor WGD on ctDNA detection. We have demonstrated that tumor WGD is associated with a higher probability of detecting ctDNA in plasma and that this association is most pronounced for low-stage disease. Knowledge about factors influencing ctDNA detection could potentially improve ctDNA detection rates and thus improve the usefulness and clinical utility of ctDNA as a biomarker. Our study paves the way for future investigations to assess the association between tumor copy numbers and ctDNA detection rates and to explore the utility of copy number integration in ctDNA detection procedures.

## Figures and Tables

**Figure 1 cancers-15-01136-f001:**
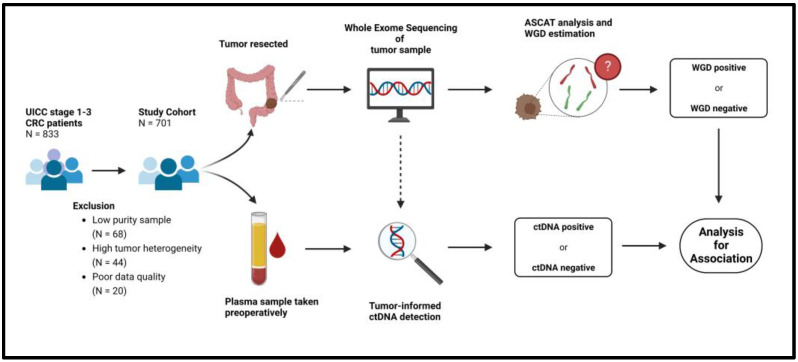
Flowchart of methods. ASCAT = Allele-Specific Copy Number Analysis of Tumors, CRC = Colorectal cancer. ctDNA = Circulating tumor DNA, WGD = Whole Genome Doubling.

**Figure 2 cancers-15-01136-f002:**
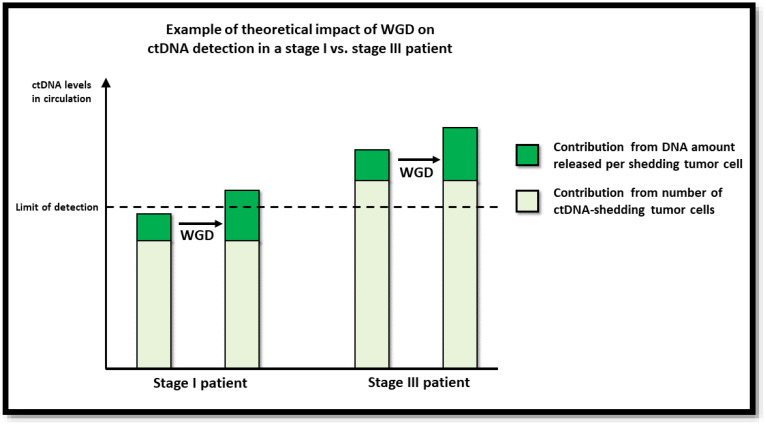
Illustration of theoretical impact of WGD on ctDNA detection.

**Table 1 cancers-15-01136-t001:** Descriptive characteristics of the included patients and the entire cohort.

Characteristic	Full Cohort, *N* = 833	Study Cohort, *N* = 701
Age—Median (Range)	71 (26, 93)	71 (26, 92)
Sex—*n* (%)		
Female	377 (45%)	318 (45%)
Male	456 (55%)	383 (55%)
UICC Stage ^1^—*n* (%)		
I	168 (20%)	141 (20%)
II	404 (48%)	336 (48%)
III	261 (31%)	224 (32%)
Tumor Location—*n* (%)		
Right Colon ^2^	400 (48%)	318 (45%)
Left Colon	258 (31%)	223 (32%)
Rectum	175 (21%)	160 (23%)
Tumor Size (mm) ^3^—Median (Range)	40 (5, 180)	40 (5, 180)
Unknown ^4^—*n*	3	3
Tumor Histological Type		
Adenocarcinoma	765 (92%)	650 (93%)
Mucinous Adenocarcinoma	62 (7%)	46 (6%)
Medullary Carcinoma	5 (<1%)	4 (<1%)
Signet Ring Cell Carcinoma	1 (<1%)	1 (<1%)
Venous Invasion—*n* (%)		
Detected	244 (30%)	204 (30%)
Not Detected	577 (70%)	486 (70%)
Unknown ^5^	12	11
MMR Status ^1^—*n* (%)		
Proficient	644 (79%)	565 (82%)
Deficient	169 (21%)	121 (18%)
Unknown	20	15
ctDNA Status ^1^—*n* (%)		
Detected	527 (63%)	444 (63%)
Not Detected	306 (37%)	257 (37%)

^1^ UICC = Union for International Cancer Control, MMR = Mismatch repair, ctDNA = Circulating tumor DNA. ^2^ Right colon defined as a tumor location proximal to left colonic flexure. ^3^ Tumor size is defined as the largest diameter measured by the pathologist in the fixated resection specimen. ^4^ In two instances, all tumor tissue was removed during the biopsy, leaving no tumor tissue to be found after resection. In these cases, the original tumor diameter could not be determined. In one instance, the tumor diameter was not assessed by the pathologist. ^5^ In 12 instances, the venous invasion was not assessed by the pathologist.

**Table 2 cancers-15-01136-t002:** Associations between tumor characteristics and ctDNA detection.

	Overall				UICC Stage I				UICC Stage II				UICC Stage III			
Characteristic	ctDNA+ ^1^ *n* = 444	ctDNA− ^1^ *n* = 257	OR ^1^ (95% CI ^1^)	*p*-Value	ctDNA+ *n* = 53	ctDNA− *n* = 88	OR (95% CI)	*p*-Value	ctDNA+ *n* = 226	ctDNA− *n* = 110	OR (95% CI)	*p*-Value	ctDNA+ *n* = 165	ctDNA− *n* = 59	OR (95% CI)	*p*-Value
**UICC Stage ^1^** —*n* (%)																
I	53 (12%)	88 (34%)	—													
II	226 (51%)	110 (43%)	3.41 (2.27, 5.16)	**<0.001**												
III	165 (37%)	59 (23%)	4.64 (2.97, 7.34)	**<0.001**												
**Tumor Location**—*n* (%)																
Right Colon	185 (58%)	133 (42%)	—		17 (32%)	37 (42%)	—		105 (46%)	68 (62%)	—		63 (38%)	28 (47%)	—	
Left Colon	152 (68%)	71 (32%)	1.54 (1.08, 2.21)	**0.019**	13 (25%)	24 (27%)	1.18 (0.48, 2.86)	0.7	79 (35%)	29 (26%)	1.76 (1.05, 3.01)	**0.034**	60 (36%)	18 (31%)	1.48 (0.75, 2.99)	0.3
Rectum	107 (67%)	53 (33%)	1.45 (0.98, 2.17)	0.066	23 (43%)	27 (31%)	1.85 (0.84, 4.17)	0.13	42 (19%)	13 (12%)	2.09 (1.07, 4.32)	**0.037**	42 (25%)	13 (22%)	1.44 (0.68, 3.16)	0.4
**Tumor Size (mm)**—Median (Range)	48 (7, 180)	30 (5, 110)	1.05 (1.04, 1.06)	**<0.001**	30 (13, 95)	20 (5, 55)	1.06 (1.03, 1.09)	**<0.001**	52 (15, 180)	35 (10, 110)	1.05 (1.04, 1.07)	**<0.001**	45 (7, 170)	33 (12, 90)	1.04 (1.02, 1.06)	**<0.001**
Unknown—*n* (%)	1	2			0	2							1	0		
**Tumor type**—*n* (%)																
Adenocarcinoma	414 (93%)	236 (92%)	—		51 (96%)	83 (94%)	—		208 (92%)	96 (87%)	—		155 (94%)	57 (97%)	—	
Other ^2^	30 (7%)	21 (8%)	0.81 (0.46, 1.47)	0.5	2 (4%)	5 (6%)	0.65 (0.09, 3.14)	0.6	18 (8%)	14 (13%)	0.59 (0.28, 1.26)	0.2	10 (6%)	2 (3%)	1.84 (0.47, 12.2)	0.4
**Venous invasion**—*n* (%)																
Not detected	291 (67%)	195 (77%)	—		45 (85%)	78 (91%)	—		169 (77%)	86 (79%)	—		77 (47%)	31 (53%)	—	
Detected	145 (33%)	59 (23%)	1.65 (1.16, 2.36)	**0.006**	8 (15%)	8 (9%)	1.73 (0.60, 5.02)	0.3	50 (23%)	23 (21%)	1.11 (0.64, 1.96)	0.7	87 (53%)	28 (47%)	1.25 (0.69, 2.28)	0.5
Unknown	8	3			0	2			7	1			1	0		
**MMR Status ^1^**—*n* (%)																
Proficient	355 (63%)	210 (37%)	—		41 (77%)	73 (85%)	—		169 (76%)	84 (79%)	—		145 (90%)	53 (93%)	—	
Deficient	81 (67%)	40 (33%)	1.20 (0.80, 1.83)	0.4	12 (23%)	13 (15%)	1.64 (0.68, 3.96)	0.3	52 (24%)	23 (21%)	1.12 (0.65, 1.99)	0.7	17 (10%)	4 (7%)	1.55 (0.55, 5.58)	0.4
Unknown	8	7			0	2			5	3			3	2		
**WGD ^1^**—*n* (%)																
No	188 (42%)	136 (53%)			18 (34%)	49 (56%)	—		104 (46%)	66 (60%)	—		66 (40%)	21 (36%)	—	
Yes	256 (58%)	121 (47%)	1.53 (1.12, 2.09)	**0.007**	35 (66%)	39 (44%)	2.44 (1.22, 5.03)	**0.013**	122 (54%)	44 (40%)	1.76 (1.11, 2.81)	**0.017**	99 (60%)	38 (64%)	0.83 (0.44, 1.53)	0.6

^1^ UICC = Union for International Cancer Control, ctDNA = Circulating tumor DNA, MMR = Mismatch Repair, WGD = Whole Genome Doubling, OR = Odds Ratio, CI = Confidence Interval. ^2^ Mucinous adenocarcinoma, medullary carcinoma, signet ring cell carcinoma.

**Table 3 cancers-15-01136-t003:** Multivariable model of the impact of tumor characteristics on ctDNA detection.

Characteristic	OR ^1,2^	95% CI ^1^	*p*-Value
Whole Genome Doubling			
No	—	—	
Yes	1.74	1.20, 2.52	**0.004**
UICC Stage ^1^			
I	—	—	
II	1.43	0.86, 2.37	0.2
III	2.23	1.29, 3.87	**0.004**
Tumor Location			
Right Colon	—	—	
Left Colon	2.14	1.37, 3.36	**<0.001**
Rectum	2.51	1.54, 4.15	**<0.001**
Tumor Type			
Adenocarcinoma	—	—	
Other ^3^	0.71	0.35, 1.45	0.3
Venous Invasion			
Not Detected	—	—	
Detected	0.95	0.62, 1.46	0.8
MMR Status			
Proficient	—	—	
Deficient	1.64	0.94, 2.89	0.082
Tumor Size (mm)	1.05	1.04, 1.06	**<0.001**

^1^ UICC = Union for International Cancer Control, OR = Odds Ratio, CI = Confidence Interval. ^2^ For categorical variables, estimates are the odds ratio of ctDNA detection for the given level compared to the reference level. For tumor size, estimates are the increase in odds ratio per mm increase in tumor diameter. ^3^ Mucinous adenocarcinoma, medullary carcinoma, signet ring cell carcinoma.

## Data Availability

All data used in this study are available in Appendix A.

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
