# Peer review of "Impact of Whole Genome Doubling on Detection of Circulating Tumor DNA in Colorectal Cancer"

_cancers, 2023, doi:10.3390/cancers15041136_

Round 1
Reviewer 1 Report
Major comments:
· What were the primary factors considered for differing sequencing depths in WES?
· How do the sequencing depths used in this study relate to those of similar published works?
· The authors use a targeted panel of genes described in Table S1. They describe these genes as being frequently mutated.
o References to back this statement would be helpful, especially for the 18 SNPs.
o Each rsID for the 18 SNPs should be added to the table
o Chromosomal coordinates would be helpful rather than “Selected frequently mutated regions”, which does not help a reader replicate this work.
· What factors may have attributed to the high loss rate of your cohort collection (~16%?).
· It is unclear whether the results described in Table 2 are as a result of association testing between only the select variables shown or following an analysis of all variables described shortly before in Table 1. Could the authors please clarify.
o Was there a difference in detection status from tumors identified in different colorectal locations? Right/Left/Rectal tumors are overlapping but distinct.
· A formulaic representation of the multivariable model used to generate results in Table 3 would be very helpful. What factors were adjusted for in these regressions?
· Figures should be presented in the Results section of a manuscript, not the discussion.
Minor comments:
· Figure S1 and S2: Authors should not use a red-green color palette. 8% of males are RG color blind.
Reviewer 2 Report
The manuscript by Kabel et al describes the analysis of the impact of whole genome doubling on ctDNA detection in CRC. At present, the pathological and biological CRC features that determine the levels of ctDNA in plasma are poorly understood. Hypothetically, it may be expected that WGD is one parameter that contributes to the levels of ctDNA. Using a large set of CRC samples, this well-written manuscript clearly addresses this interesting question.
Several issues remain to be addressed:
· MSI status is not taken along. The biology of MSI tumors is quite different from microsatellite stable (MSS) tumors. What is the impact of WGD-status on ctDNA detection in MSS CRC compared to all CRC?
· A binary outcome is presented, test results being either ctDNA-negative or ctDNA-positive. This is related to the limit of detection of the ctDNA-detection technology that is used, and may explain why WGD does not increase ctDNA detection in stage III CRC in this study. Can ctDNA levels be represented more quantitatively? In that case, what is the association between WGD-status and ctDNA levels in stage III CRC? And in CRC overall?
· Emerging technologies in the field of ctDNA detection, such as whole genome sequencing-based approaches, will increase the sensitivity of ctDNA testing. Briefly discuss your WGD-findings in context of these technological developments for the clinical setting.
* Please check readability of lines 278/279: “The association between persisted after adjusting for other tumor characteristics.”
Reviewer 3 Report
Reviewer´s comments:
Whole genome doubling (WGD) is a phenomenon known to the scientific community for a long time, though in the time of improvements in sequencing technologies its presence has reemerged. Authors are presenting their research based on WGD assessment in colorectal tumor samples and its association with the possibility to detect ctDNA in plasma. The basic assumption is that tumors with WGD+ will lead to higher ctDNA level in plasma.
Methods and Materials:
It is well described. Authors have collected tumor tissue samples (either FF or FFPE) and corresponding blood from 833 patients. Basic characteristics are provided in Table 1 of the Manuscript. Based on the inclusion criteria the total number decreased to 701 samples due to:
1. Impossibility to detect cfDNA,
2. low quality WES data.
It is speculative, if presentation of descriptive characteristics in Table 1 of the previous 833 patients is necessary, although I understand that you wanted to show that the final cohort was indifferent from the original one.
WES data of 701 tumor DNA and corresponding 701 PBMC DNA were evaluated by appropriate bioinformatic analysis tools to provide information about:
1. Presence of tumor-specific mutations (GATK Mutect2 somatic variant calling)
2. Validation of appropriate sample pair (tumor DNA matched blood DNA based on SNP evaluation)
These data were further used for the determination of which assay will be applied for the ctDNA detection. ddPCR and deep seq are both highly sensitive approaches for the detection of minimal amount of ctDNA. Individuals harbouring mutation present in the in-house list of ddPCR assays were preferentially analyzed by ddPCR method. Other samples (lesser amount) were analyzed by a customized targeted NGS panel (12 genes and 18 SNPs) for deep sequencing (median coverage 15716x).
1. By ddPCR method a specific setup was applied. I don´t understand the concept of 6 reactions per each sample where in the description you are missing the cfDNA sample. In the lines 173-176 your description mentions:
a. No template control
b. Tumor DNA positive sample
c. PBMC negative control
d. PBMC negative control
e. Patient´s own PBMC DNA
f. PBMC from a healthy donor
2. What is the difference between c/d and f?
3. Are the 6 cfDNA sample aliquots mentined in line 172 missing in the further description?
WDG was assessed by ASCAT method described in a dedicated paragraph with citing of appropriate literature.
Result section:
In Figure 1 – The scheme is not precise, it does not reflect what is written in the description of Methods and Materials. Please re-do this nice figure. Tumor DNA sample did not undergo Deep-targeted NGS.
Table 2 – Provides data summary for association study between several factors. The information is clear.
Why did you not cover the information of the localization of the tumor in association with WGD detection?
Based on the data, WGD rate was the highest for ctDNA positive stage I patients (66%). Interestingly, while WGD status impacted ctDNA detection in stage I and stage II patients, it did not in stage III patients, although patients with late stage cancer often have more ctDNA in the circulation. Authors provide explanation for such an occurrence by WGD not being the major phenomenon in late stage disease in opposite to factors mentioned in lines 283-287. If so, there should be higher frequence of ctDNA capture in samples from late-stage CRC (165 ctDNA+:59 ctDNA-). Authors are aware of this and mention it as a limitation of this study, that not all of the samples were captured as ctDNA+. Since ddPCR and deep-seq NGC methods are the most appropriate approaches with high sensitivity for samples with limited number of variant allele presence, I find this explanation satisfactory, although hard to accept.
According to obtained data, WGD is not informative for the decision if ctDNA will be detected in plasma sample of a patient.
One of the factors that could possibly affect the presence or absence of ctDNA in the plasma of tumor-affected patients could be the genetic signature – specific driver mutations presence. I would be very curious to evaluate your data even more and look for association of WGD and ctDNA presence after stratification based on TP53 mutation presence, for example. Your discussion section should, to my opinion, be enriched by information from this field since there are already some publications dealing with this topic.
For example:
https://doi.org/10.1038/s12276-021-00583-1
High prevalence of TP53 loss and whole-genome doubling in early-onset colorectal cancer
https://doi.org/10.1038/s41467-022-31899-9 /
Oncogenic BRAF induces whole-genome doubling through suppression of cytokinesis
Reference section:
Authors have well chosen the cited literature.
